

# Control Co-Design optimization of floating offshore wind turbines with tuned liquid multi-column dampers

Wei Yu[1], Sheng Tao Zhou[2], Frank Lemmer[3], and Po Wen Cheng[1]

[1]Stuttgart Wind Energy (SWE) at Institute of Aircraft Design, University of Stuttgart
[2]Powerchina Zhongnan Engineering Co., Ltd
[3]sowento GmbH

**Correspondence:** Wei Yu (yu@ifb.uni-stuttgart.de)

**Abstract.** The technical progress in the development and industrialization of Floating Offshore Wind Turbines (FOWTs) over the past decade is significant. Yet, the higher Levelized Cost of Energy of FOWT, compared to onshore wind turbines, is still limiting the market share. One of the reasons for this is the larger motions and loads caused by the rough environmental excitations. Many prototype projects tend to employ more conservative substructure designs to meet the requirements on motion

dynamics and structural safety. Another challenge lies in the multidisciplinary nature of a FOWT system, which consists of several strongly coupled subsystems. If these subsystems cannot work in synergy with each other, the overall system performance may not be optimized. Previous research has shown that a well-designed blade pitch controller is able to reduce the motions and structural loads of FOWTs. Nevertheless, due to the negative aerodynamic damping effect, improving the performance by tuning the controller is limited. One of the solutions is adding Tuned Liquid Multi-Column Damper Dampers (TLMCDs), meaning a structural solution to mitigate this limiting factor for the controller performance. It has been found that

the additional damping, provided by TLMCDs, is able to improve the platform pitch stability, which allows a larger blade pitch controller bandwidth and thus a better dynamic response. However, if a TLMCD is not designed by taking the whole FOWT system dynamics into account, it may even deteriorate the overall performance. Essentially, an integrated optimization of these subsystems is needed. This paper has developed a Control Co-Design optimization framework for FOWTs installed with TLMCDs. By using the multi-objective optimizer Non-Dominated Sorting Genetic Algorithm II, the objective is to optimize

the platform, the blade pitch controller and the TLMCD simultaneously. Five free variables characterizing these subsystems are selected and the objective function includes the FOWT's volume of displaced water (displacement), several motion and load indicators. Instead of searching for a unique optimal design, an optimal Pareto surface of the defined objectives is determined. It has been found that the optimization is able to improve the dynamic performance of the FOWT, quantified by motions and

loads, when the displacement remains similar. On the other hand, if motions and loads are constant, the displacement of the FOWT can be reduced, which is an important indication of lower manufacturing, transportation and installation costs. In conclusion, this work demonstrates the potential of advanced technologies such as TLMCDs to advance FOWTs for commercial competitiveness.



## 1 Introduction

Structural control techniques play an important role in mitigating undesired motions or loads across various disciplines. For Floating Offshore Wind Turbines (FOWTs), the implementation of Tuned Mass Dampers (TMDs) or Tuned Liquid Column Dampers (TLCDs) has been extensively investigated over the past years (Rotea et al., 2010; Lackner and Rotea, 2011; Cross-Whiter et al., 2018; Tong et al., 2018). These devices, when strategically installed within the nacelle or substructure, offer promising solutions in reducing structural vibrations and enhancing overall system stability. The specific geometric characteristics of FOWTs, as well as the requirement for bi-directional damping effects, necessitate the exploration of novel damping mechanisms. As a result, Tuned Liquid Multi-Column Dampers (TLMCDs) have been recommended for semi-submersible or barge-type FOWTs (Coudurier et al., 2018), demonstrating the potential to effectively mitigate the dynamic responses of FOWTs. This numerical model for TLMCDs is further coupled to an aero-hydro-servo-elastic tool for FOWTs in Yu et al. (2023), and validated by comparison with experimental results.

Nevertheless, the application of TLMCDs alone may not fully address the unique challenges posed by FOWTs. The complex hydrodynamic interactions, coupled with the aerodynamics and servo dynamics, have a profound impact on the effectiveness of this technology. This makes it difficult to make progress by focusing solely on the design of TLMCDs without considering their impact on other subsystems and the overall system of a FOWT. In particular, a FOWT is an actively controlled system, where dynamic characteristics play a central role in determining the design of the control system. By providing additional damping, TLMCDs increase the flexibility to achieve effective and robust control for FOWTs. Taking into account all these highly coupled subsystems, the question arises as to how the benefits of TLMCDs can be exploited and the overall system behaviour can be optimized.

Given these challenges, integrated optimization techniques are the most suited solutions to this problem. By exploring the design space of the whole FOWT system, design parameters that provide the best synergy between the coupled subsystems can be found. General best practices for design optimization of FOWTs have been extensively addressed in previous research. The aim of such optimization work is to reduce the Levelized Cost of Energy (LCOE) of FOWTs. For the research community, this is usually represented by a reduction in size and weight, motion and loads. A lot of work has been done on design optimization. (Hall et al., 2013) has applied genetic algorithm to optimizes the hull shape and mooring system with both single- and multi-objectives. The fitness of the three substructure concepts on their Pareto fronts was evaluated and compared. The selected design parameters characterizing the hull geometry and mooring layout are the optimal choices for each concept, ensuring a fair comparison. As the blade pitch controller interacts with the dynamics of the floating platform and instabilities may occur, (Lemmer et al., 2017b) has presented an integrated substructure optimization including a self-tuning controller. The work has been further extended in (Zhou et al., 2021) by adding the mooring system using a multi-objective optimizer, showing the significant impact of the mooring system on the optimal solutions. More comprehensively, (Hegseth et al., 2020) established an integrated design optimization framework, in which the platform, tower, mooring system and blade pitch controller are





optimized simultaneously by using a gradient-based optimizer, considering both fatigue and extreme response constraints. A weighted combination of system cost and power quality is employed as the objective function, resulting in an unconventional hourglass shaped spar buoy floater. More recent publications have also highlighted the importance of control co-design (CCD).

Since the control plays an important role in terms of motions and loads, it is important to optimize the blade pitch controller simultaneously during the design process. The ARPA-E ATLANTIS Program [2] has already announced several projects focusing on this topic (Garcia-Sanz, 2019).

Building on these prior research and findings, this work explores the limits of the TLMCD's contribution to mitigating motions and loads of the FOWT system. This is done through a multi-objective optimization approach that incorporates Control

Co-Design (CCD) to simultaneously optimize the design of the substructure, the TLMCD, and the blade pitch controller. By coordinating the function of these subsystems, the framework is expected to achieve the best possible synergy to maximize the potential of TLMCDs in improving the performance of FOWTs at a systems engineering level.

## 2 Impact of the TLMCD and the controller

As emphasized in the previous introduction, the dynamic responses of FOWTs are significantly affected by the blade pitch

controller and the TLMCD. In this section, the impacts of these influential subsystems are visually demonstrated, which provides a comprehensive understanding of these effects.

### 2.1 Reference FOWT and numerical tools

The original NAUTILUS-DTU10 MW FOWT (Yu et al., 2018b) is investigated as a case study, which will be referred to as NAUTILUS-10 in the following discussion. The installation sketch is shown in Figure 1. The analysis involves analyzing

the linearized system and performing coupled time simulations. The linear analysis focuses on the influence of the controller and the TLMCD on the stability margin and step response to wind of the coupled system, while the coupled simulations will evaluate the system performance in more realistic operation conditions. As for the numerical tool, Simplified Low Order Wind Turbine model (SLOW) is used for both linear analysis and coupled simulation, which is originally developed in (Lemmer et al., 2020) and coupled to the TLMCD model in (Yu et al., 2023).

### 80 2.2 Linear analysis

For the stability margin, the Nyquist plot is used to visualize how system stability changes with control designs. Figure 2 presents the Nyquist plot at an operating wind speed of $16\,\mathrm{m/s}$ with various controller gains and TLMCD setup. The integral time constant $T_i$ of the PI controller is kept constant, while different proportional gains are selected. In addition, the two systems with and without a TLMCD are compared, i.e. a reference system without TLMCD in the subplot on the left and

on the right the FOWT is stabilized by a TLMCD system. The highlighted point (-1, 0) in red represents the unstable point. According to the stability criteria using the Nyquist plot, the system becomes unstable when the contour lines encircles this

---

[2]https://arpa-e.energy.gov/technologies/programs/atlantis. Accessed on 21.June.2023





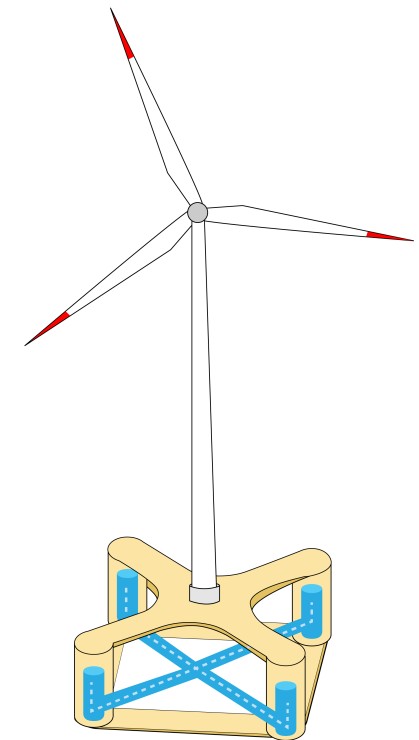

**Figure 1.** Design and installation of the TLMCDs for the NAUTILUS-DTU10MW FOWT.

point. When the proportional gain $k_p$ increases, the contour lines get closer to the unstable point and eventually the system becomes unstable. The shortest distance between (-1, 0) and the contour line $dist_s$ determines the stability margin, quantified by the sensitivity peak $M_s = 1/dist_s$. The larger the distance, the more robust the control system becomes. This allows a higher

uncertainty in the numerical modelling. By comparing the two subplots, the contribution of a TLMCD is also visible. The contour lines are moved further away from the unstable point with the same control gains. This proves that the TLMCD can improve the dynamic behavior of the system by increasing the stability margin.

In addition to the stability margin, the step responses are also an important measure to describe the dynamic behavior. How the step responses to wind change with control gains and TLMCD setups is presented in Figure 3. Several standard definitions

in control theory are used to quantify the response behaviour, including the rise time $T_r$, the settling time $T_s$ and the overshoot, which are described by (Yu et al., 2020) in detail. As shown, when $T_i$ is constant, a relatively large $k_p$ improves the disturbance rejection, implied by smaller overshoot and shorter rise time. However, this also leads to a negative effect on the settling time. With a higher gain it takes longer for the system to reach a steady state after a step disturbance, which is equivalent to the situation when a system has insufficient damping. In some extreme cases, the system can not converge any more. By comparing

the above and below subplots of Figure 3, the impact of the TLMCD can be inferred. First of all, the TLMCD does not change the response to the step wind within around $40\,\mathrm{s}$. This is mainly dominated by the aerodynamics, since the turbine and control

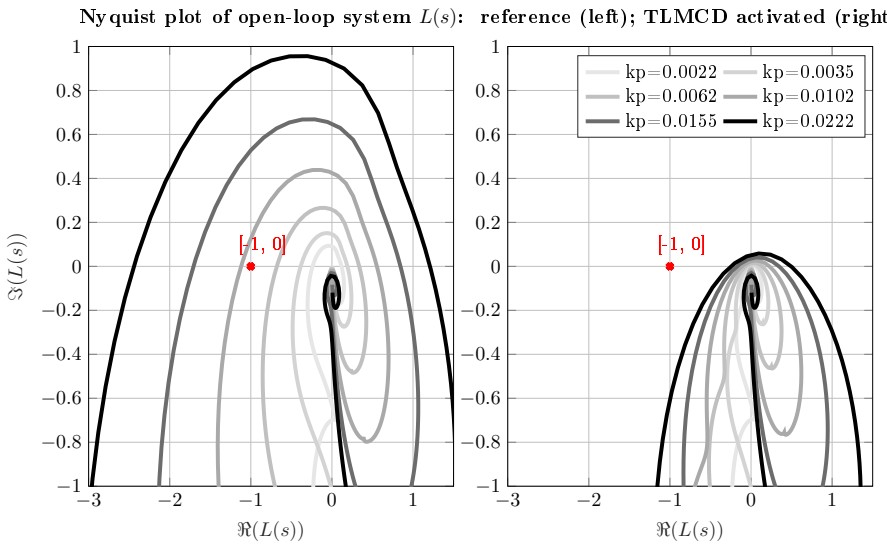

**Figure 2.** Nyquist plot of open loop transfer function $L(s)$: NAUTILUS-DTU10 MW FOWT with fixed time constant $T_i$ ($T_i$=8 s) and varying proportional gain $k_p$ (value increases when the darkness increases) at wind speed 16 m/s.

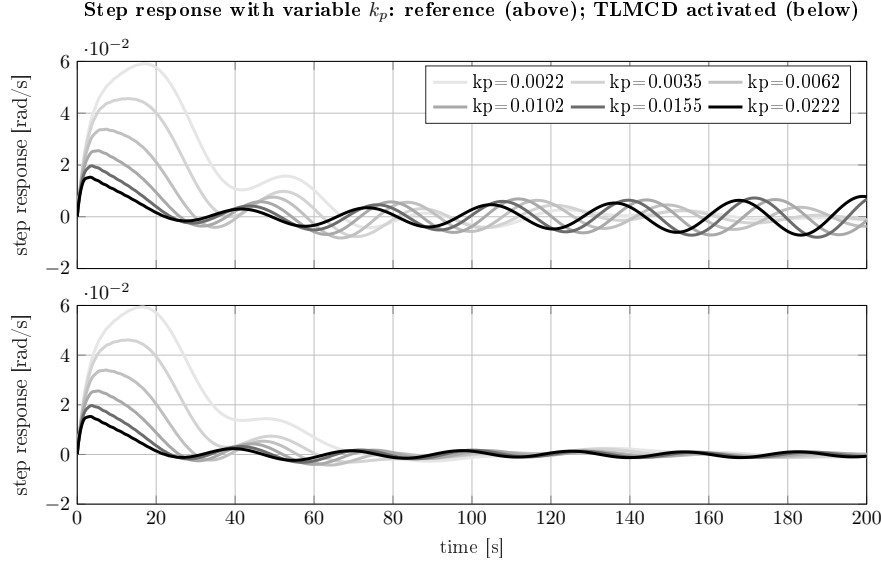

**Figure 3.** Step response of generator speed to unit step wind with different control gains at 16 m/s, $k_p$ increases with the color darkness increases.





parameters are the same, one can expect the same rise time and overshoot. After this period, the platform motion continues to oscillate due to the coupled dynamics, resulting in a longer settling time than onshore turbines. When the TLMCD is activated, the low damping caused by the higher $k_p$ is partially compensated, so that the coupled dynamics of the platform motions are better damped.

Combining the observations on the sensitivity margin in Figure 2, it can be concluded that an improvement in generator speed control can be achieved at the expense of stability. This is also reported in (Yu et al., 2018a) as a trade-off between the generator speed regulation and platform pitch motion, which is found by studying a 5 MW FOWT. While a TLMCD can break this limitation by introducing additional damping into the system, a more aggressive controller with higher gains is generally possible. This not not improves the disturbance rejection, but also maintain the system stability.

## 2.3 Coupled simulation

The linear analysis clearly demonstrates the significant influence of both the TLMCD and the blade pitch controller on the system stability, as well as their role in shaping the closed-loop dynamic responses. However, it is still unclear how these subsystems interact and affect the coupled dynamic responses in real operating conditions. In order to gain a deeper insight into the coupled dynamics, a more comprehensive study using coupled aero-hydro-servo-elastic time simulations is conducted, and the statistics of the simulation results are presented in Figure 4. Here the original NAUTILUS-10 design is used as a reference for simulations to compare three different setups. These setups include scenarios with the TLMCD activated or deactivated, and the controller in its original state or redesigned. The original controller, as designed in (Lemmer (né Sandner) et al., 2019), prioritizes stability margins without considering the presence of the TLMCD. In contrast, the redesigned controller takes into account the additional damping from the TLMCD and shapes the controller behaviour through step responses under the stability constraints, the resign procedure is described in (Yu et al., 2020).

The first step is to understand the impact of the TLMCD. In both cases, i.e. with the reference controller and with the redesigned controller, the TLMCD can improve the pitch motion and the tower base bending moment, more significantly at higher wind speeds. However, this improvement, especially the rotor speed performance, is independent of the blade pitch controller. When the TLMCD is active, redesigning the controller can significantly improve the rotor speed performance, evident by comparing the solid yellow and black bars. As the generator torque is constant in the simulation, the rotor speed also represents the power production quality.

The impact of the TLMCD is examined as a first step. In both cases, i.e. with the reference controller and with the redesigned controller, the TLMCD improves pitch motion and tower base bending moments, particularly at higher wind speeds. However, the rotor speed performance, crucial for power production quality, depends on the blade pitch controller. When the TLMCD is active, redesigning the controller significantly enhances rotor speed performance. As the generator torque is constant in the simulation, the rotor speed also represents the power production quality.

In summary, adding structural damping to an actively controlled system does not automatically guarantee improved overall system performance. It is crucial to optimize the controller in conjunction with the damping system to achieve synergy and maximize benefits. This finding inspires the work in the next section, where CCD optimization techniques are employed. By

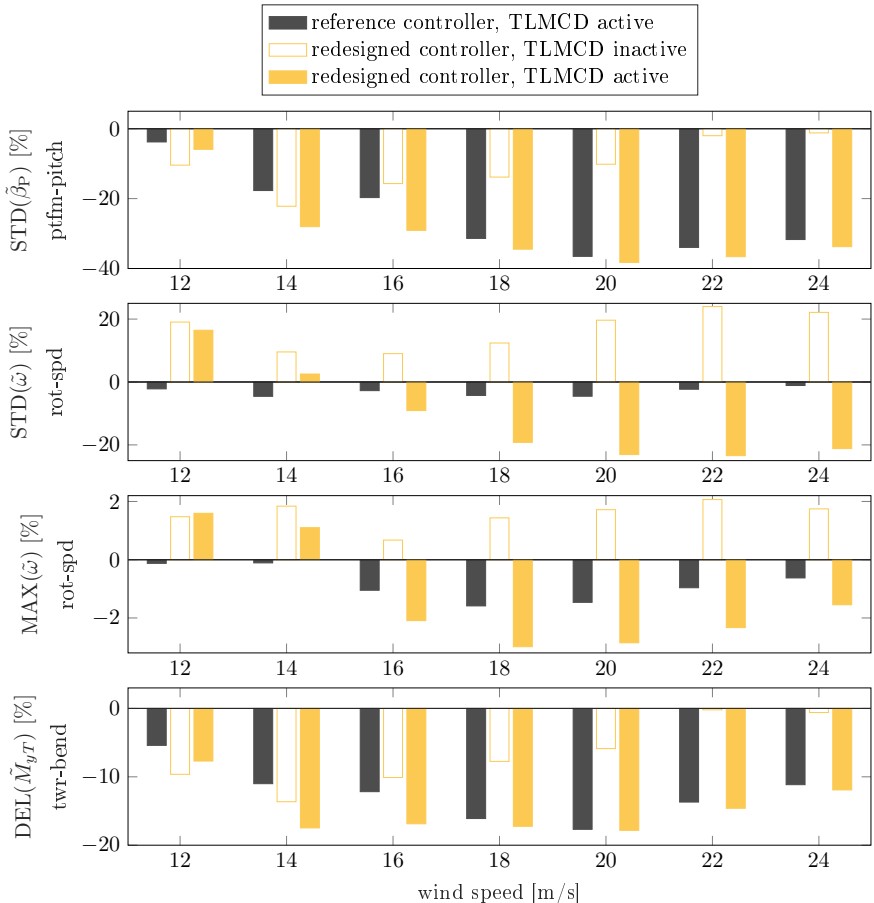

**Figure 4.** Comparison of relative system statistical responses of the NAUTILUS-DTU10MW FOWT w.r.t. the case with a reference controller and without a TLMCD at different operating wind speeds (sea states based on the design load cases of a site in the LIFES50+ project).

systematically considering the effects of the substructure design, the additional damping, and the active control system, the overall performance of the system is expected to be maximized.

## 3 Optimization Framework

This section outlines the setup of the optimization framework, which involves design space, optimizer, constraints and cost
140 model. These elements are essential to ensure that the optimization process is well-defined and that all subsystems can be optimized to maximize the desired performance while staying within the constraints. The entire framework is implemented in MATLAB®, using the linearized SLOW for controller design and nonlinear SLOW for coupled simulation.





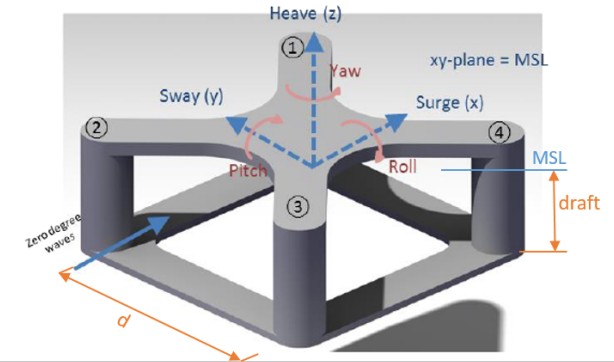

**Figure 5.** Illustration of the design variables of the substructure used for the optimization (Yu et al., 2018b)(Zhou et al., 2019).

## 3.1 Design space

The optimization framework for the FOWT system includes several subsystems, i.e. the floating platform, the blade pitch controller, and the TLMCD. To ensure a manageable computational complexity, it is beneficial to limit the number of free variables for each subsystem. For the platform, the design space is derived from (Zhou et al., 2019), with the free design variables comprising of the column spacing to the center line $d$ and the draft of the platform. These variables were chosen based on a sensitivity study conducted during the LIFES50+ project (Lemmer et al., 2017a), which has highlighted their significant impact on the dynamic responses. Although the column diameter was also found to be an influential variable in dynamics responses (Zhou et al., 2021), only two design variables are selected for the optimization due to limited computational resources. These variables are highlighted in orange in Figure 5, while the remaining dimensional parameters stay consistent with the original NAUTILUS-10 design (Yu et al., 2018b).

The fairleads of the mooring system are attached to the outer walls of the four vertical columns. As the column spacing is a free variable, the fairleads' positions in the body frame of the floater are affected by the column spacing $d$. Therefore, the radius of the fairleads can be expressed as a function of $d$. This means that changes in $d$ can lead to corresponding changes in the fairlead radius, which is expressed as:

$$r_{\text{fairlead}} = 5.25 + \frac{\sqrt{2}}{2}d \tag{1}$$

where $5.25$ is the original radius of the pontoon taken from (Yu et al., 2018b), in meters.

Regarding the controller design, the automated design procedure in (Yu et al., 2020) is used. It was found that the design parameter rise time $T_r$, which shapes the closed loop step response, has a significant impact on the dynamic behavior of the controller design. As the linear model used for the closed loop analysis varies with the operating wind speeds, it would be ideal to optimize $T_r$ for the entire range of operating mean wind speeds. However, this leads to an extremely large design space. To simplify the optimization, only two free variables are chosen, i.e. the rise time at operating wind speeds of $12\,\text{m/s}$ and $24\,\text{m/s}$. For other operating points, it is assumed that $\frac{1}{T_r}$ increases linearly with wind speed, allowing a smooth transition





**Table 1.** Free variables defined for the optimization process.

| Property | Unit | Range | Min scale |
|---|---|---|---|
| Column spacing $d$ | m | [35, 70] | 2 |
| Platform draft | m | [10, 64] | 2 |
| TLMCD head loss $\eta$ | - | [4, 9] | 1 |
| Relative rise time $T_{r,12}$ | % | [80, 110] | 5 |
| Relative rise time $T_{r,24}$ | % | [25, 60 ] | 5 |

between different operating points. Once the values of $T_r$ are determined, the control parameters, the proportional gain $k_p$ and
the integral gain $k_i$ can be determined. It should be noted that the choice of $T_r$ is influenced by the natural frequency of the
platform pitch motion. As a result, the values of $T_r$ are expressed as relative values with respect to the natural frequency of the
platform pitch, expressed as a percentage of the natural frequency. The ranges for these percentages are narrowed down by a
prior brute force study.

As for the TLMCD, its horizontal arm must have a length of $\sqrt{2}d$ to accommodate the vertical columns of the TLMCD in
the floater's vertical pontoons. The height of the vertical columns remains at $70\,\%$ of the draft. This leaves only two geometric
free variables of the TLMCD left, which are determined by setting the TLMCD fluid mass and the natural frequency. A lower
TLMCD mass is generally favourable for the static stability, but generates limited stabilizing moment. It is suggested in (Gawad
et al., 2001) that the best choice of TLCD mass ratio for ships is around $3.5\,\%$, so the TLMCD mass ratio of $3\,\%$ is chosen
in this work. The natural frequency of the TLMCD is equal to that of the platform pitch. As a result, only the head loss $\eta$ is
defined as a free design variable, which is a measure of energy loss of the TLMCD fluid system. On the one hand, it is an
important factor for the floater dynamics, on the other hand, $\eta$ can be adjusted by adding baffles inside the TLMCD, another
reason for its inclusion as a free variable.

Table 1 summarizes all five free variables that are considered in the optimization. To accelerate the optimization process,
minimum steps have been defined for each variable, which results in the design space being discretized instead of continuous.
This means that the variables can only take on certain values within their respective ranges, rather than any possible value
within the range. By discretizing the design space, the optimization process can be performed more efficiently, which can help
to reduce the time and computational resources required for the optimization.

### 3.2 Multi-objective optimizer

When evaluating the system performance of FOWTs, various factors come into play to define the cost model and assess the
effectiveness of the FOWT design. These factors include structural loads, energy production, motions, and more. To consolidate
these factors into a single objective function, weight coefficients are generally used to obtain the optimal solution. However,
determining these coefficients poses challenges, especially in academic research settings where input from industry experts on
realistic weighting factors may not be available. Furthermore, the choice of weight coefficients has a significant impact on the





optimization process and final outcomes. Recognizing this, a multi-objective optimizer is chosen to address these complexities. By employing a multi-objective optimizer, designers can obtain a range of favorable designs across different scenarios, rather than a single unique optimum, which offers valuable flexibility and provides designers with a comprehensive understanding of trade-offs and design considerations.

Therefore, the NSGA-II (Non-dominated Sorting Genetic Algorithm II) is selected in this study. A comprehensive explana-
tion of the algorithm can be found in (Deb et al., 2002). Regarding the general setup of NSGA-II, the initial population size is determined based on a general rule of thumb (Storn, 1996), which suggests a population size of approximately 10 times the number of design variables. Two stopping criteria are implemented, namely the maximum allowed number of generations and the Mutual Domination Rate (MDR), which is used to quantify the improvement of each generation (Martí et al., 2016).

### 3.3 Cost model

The cost model is an essential part of the optimization framework that can significantly influence the optimization results. While minimal LCOE is generally accepted as a good objective function in the wind industry, it is derived from a wide range of factors, some of which are not relevant to the subsystems under investigation, such as policy, market, or supply chain-related issues. Moreover, certain components of the LCOE may vary in different markets or change with suppliers, making it less informative and potentially unable to reveal the influence of the design parameters. As a result, indicators that not only reflect
the LCOE but also have physical meanings and strong correlations with the design variables are used for optimization purposes.

Three indicators are selected for the optimization: a motion indicator, a load indicator, and a cost indicator. The motion indicator will be measured using sensors for the platform pitch and nacelle fore-aft acceleration, while the load indicator will be determined by measuring the tower base bending moment and mooring fairlead tension. Instead of calculating the actual cost for materials, manufacturing, transportation, and so on, the total displaced tonnage (i.e., the weight of water displaced by
the FOWT in normal operation) is used as the cost indicator. Although the term "tonnage" can have different meanings in the shipping industry, depending on the loading condition of the vessel, the displaced tonnage is an important measure that can provide a qualitative indication of manufacturing, operation, and maintenance costs. Unlike direct cost calculation, which has many uncertainties and can vary over time and markets, displaced tonnage is a physical value that can be accurately calculated from the structural model. Therefore, it is the only measure used here to indirectly represent all costs associated with material,
manufacture, transportation, and installation.

To account for the different units and magnitudes of the selected indicators, they are normalized by comparison to the original NAUTILUS-10 design. Denoting the displacement, the Damage Equivalent Load (DEL) of the tower base bending and the mooring fairlead tension, the STD of the platform pitch motion and the tower top acceleration as $V_{\text{disp}}$, $\text{DEL}_{\text{MyT}}$, $\text{DEL}_{\text{moor}}$, $\text{STD}_\beta$, $\text{STD}_{\text{TT}}$, respectively. By compare to these values of the original NAUTILUS-10 design, indicated by the



subscript "0", the objective functions at each wind speed $\bar{u}_i$ can be expressed as:

$$
\begin{aligned}
J_1 &= \frac{V_{\text{disp, child}} - V_{\text{disp,0}}}{V_{\text{disp,0}}} \\
J_2(\bar{u}_i) &= \frac{\text{DEL}_{\text{MyT, child}} - \text{DEL}_{\text{MyT,0}}}{2 \cdot \text{DEL}_{\text{MyT,0}}} + \frac{\text{DEL}_{\text{moor, child}} - \text{DEL}_{\text{moor,0}}}{2 \cdot \text{DEL}_{\text{moor,0}}} \\
J_3(\bar{u}_i) &= \frac{\text{STD}_{\beta, \text{child}} - \text{STD}_{\beta,0}}{2 \cdot \text{STD}_{\beta,0}} + \frac{\text{STD}_{\text{TT, child}} - \text{STD}_{\text{TT,0}}}{2 \cdot \text{STD}_{\text{TT,0}}}.
\end{aligned}
\tag{2}
$$

As can be seen, both $J_2$ and $J_3$ show variations over different wind speeds. In commercial applications, it is ideal to weigh these objectives according to the probabilistic distribution of wind speeds. However, the resulting conclusion will inevitably depend on the chosen wind distribution. Since this study focuses on the general methodology rather than deriving an optimal industrial design, the objective functions for each mean wind speed are simply averaged, resulting in a final cost function with multiple objectives:

$$
\begin{aligned}
J &= [\bar{J}_1, \bar{J}_2(:), \bar{J}_3(:)] \\
\bar{J}_i(:) &= \text{mean}(J_i(\bar{u}_1), \ldots, J_i(\bar{u}_n)), \quad i = 2, 3.
\end{aligned}
\tag{3}
$$

### 3.4 Constraints

In order to accelerate the optimization process, a set of design constraints have been defined to eliminate unfeasible designs. These constraints can be classified into two type: static and dynamic. The static constraints are applied at the beginning of the optimization process and immediately reject any designs that fail to meet the requirements, thus reducing the number of designs to be simulated. The dynamic constraints are applied during the simulation and take into account the behavior of the designs under various load conditions.

The static constraints are checked before any computationally intensive time simulation is conducted. If an individual design fails to satisfy the constraint criteria, it will be excluded from further evaluation. The algorithm will continue to search for potential candidates in order to maintain the size of the design candidates to be evaluated. The constraints on the natural frequencies of the floater are primarily intended to avoid the wave frequency range. These static constraints are defined as follows:

- The displaced tonnage should not be more than twice as much as that of the NAUTILUS-10 design, i.e. $J_1 <= 1$.

- The maximum static pitch angle should be smaller than $10 \deg$.

- The heave natural period should be greater than $15 \, \text{s}$.

- The pitch natural period should be greater than $18 \, \text{s}$.

To further refine the optimization process, dynamic constraints are defined based on statistical analysis of dynamic simulations. If a design exceeds these constraints, the algorithm sets high values to the cost model $J$, redirecting the optimization towards alternative designs. The dynamic constraints consider various scenarios, including:





- – The generator overshoot should be less than $30\,\%$.

- – The maximum dynamic pitch should not exceed $12\,\mathrm{deg}$.

- – The nacelle acceleration should be smaller than $0.3\,g$, where $g$ is the gravitational acceleration.

It is worth noting that Ultimate Limit State (ULS) and Fatigue Limit State (FLS) are not considered. The selection of
constraints is based more on experience and established rules of thumb derived from previous research projects. This approach
not only guarantees timely convergence, but also allows a wider range of potential solutions to be explored within the design
space. However, it is important to emphasize that the values chosen may be conservative and may not necessarily align with
commercial standards.

### 3.5 Workflow for Objective Evaluation

The most important and computationally intensive step is the performance evaluation based on the objective functions, which
includes the design and modeling of the subsystems, the execution of coupled simulations in the time domain, as well as
the post-processing of the simulation data to compute the objective functions. Figure 6 illustrates how these processes are
interconnected during the the evaluation of the possible designs. It can be roughly divided into three parts: the preparation of
the dynamic model inputs (in blue), the model linearization and controller design (in orange), and the coupled design load case
simulation (in yellow).

#### 3.5.1 Inputs preparation of the dynamic model

The substructure design module is the first step in the optimization process. It takes the design variables of each offspring
created by the generic algorithm as input and calculates the inertial properties of the FOWT based on its geometrical variables.
At the same time, the module generates a mesh for the wet surface of the substructure and associated panel coordinates. The
data produced here are then passed on to the hydrodynamic module. The main function of this module is to generate the
hydrodynamic coefficients using the panel code ANSYS-AQWA. In addition, the module calculates the Response Amplitude
Operator (RAO) and natural frequency of the platform pitch motion, which serve as input data for the TLMCD design module.
These design modules are developed in (Lemmer et al., 2017b) and also utilized in (Zhou et al., 2019). The TLMCD design
module uses the same method as that presented in (Yu et al., 2019). The main objective is to ensure that the TLMCD has the
same natural frequency as the platform pitch, while also keeping the total fluid mass within the TLMCD constant at $3\,\%$ of the
total FOWT mass.

#### 3.5.2 Model linearization and controller design

After all inputs for the dynamic plant are set up, steady states for various operating wind speeds can be simulated and calculated.
These steady states are then used to linearize the model. The linearization of the FOWT is introduced in (Lemmer et al., 2020),
while the linerization with respect to the TLMCD and the coupled TLMCD and FOWT system is established in (?). For the





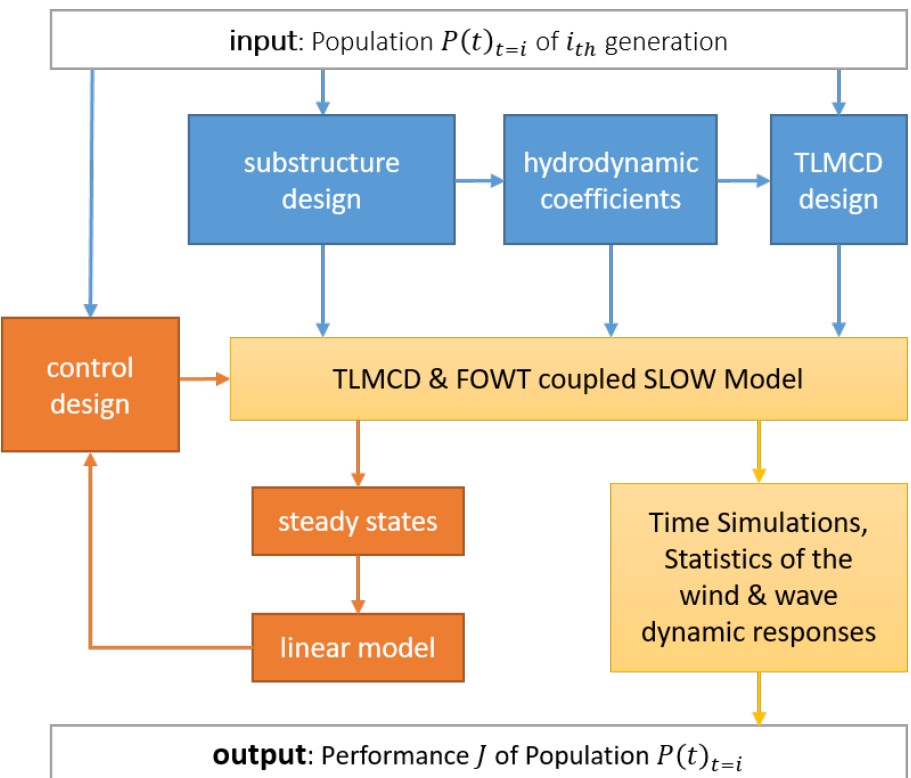

**Figure 6.** Workflow of evaluating offspring of the $i_{th}$ population.

**Table 2.** Design load cases used for optimization.

| Significant wave height $H_s$ [m] | 1.38 | 1.67 | 2.2 | 3.04 | 4.29 | 6.2 | 8.31 |
|---|---|---|---|---|---|---|---|
| Wave peak period $T_p$ [s] | 7 | 8 | 8 | 9.5 | 10 | 12.5 | 12 |
| Mean wind speed $\bar{u}$ [m/s] | 5 | 7.1 | 10.3 | 13.9 | 17.9 | 22.1 | 25 |

automated control design methodology, a comprehensive study is presented in (Yu et al., 2020). The only difference here is that the design approach targets a minimum overshoot within the feasible design space, rather than a minimum settling time as in (Yu et al., 2020).

### 3.5.3 Coupled design load case simulation and cost evaluation

Once the dynamic plant and controller are established, the final step is to perform coupled time domain simulations using a subset of design load cases recommended in the LIFE50+ project (Krieger et al., 2015). These load cases are listed in Table 2 and are used to evaluate the performance indicators for the objective functions.





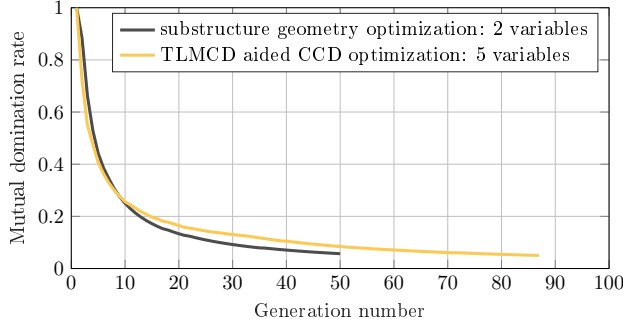

**Figure 7.** Mutual domination rate of the optimization process showing the convergence of the optimization process.

## 4   Optimization Results

This section discusses the results of two rounds of optimization performed to determine optimal designs. The first round only
considers the geometric optimization of the substructure using two design variables, i.e. column spacing $d$ and substructure
draft. In the second round, all five variables listed in Table 1 are used in the optimization process that involves all subsystems.
The results of both rounds are detailed in this section.

### 4.1   Initialization and convergence

For the first round of optimization, where only the substructure is optimized, a population size of 20 is used. For the second
round, which involves the TLMCD and blade pitch controller in the optimization loop, the population size is increased to 50.
Checking for convergence is essential to demonstrate the validity of the optimal solutions generated by the optimizer when
using genetic algorithms. In this respect, Figure 7 presents the convergence progress of both optimization rounds, indicating the
evolution of the two stopping criteria previously defined. For the substructure-only optimization, it stops when the maximum
generation of 50 is reached, and the corresponding MDR value is 0.06. However, for the TLMCD-assisted CCD optimization,
the optimization process stops before the 90th generation, as the MDR value reaches the stopping threshold of 0.05, rather than
reaching the maximum allowed 100 generations. These results show the effectiveness of the two stopping criteria in ensuring
convergence in the optimization process.

### 4.2   Optimal objective space

As mentioned earlier, the optimization process involves two rounds with different objectives. The following section discusses
how the definition of the objectives can influence the final optimal solution.



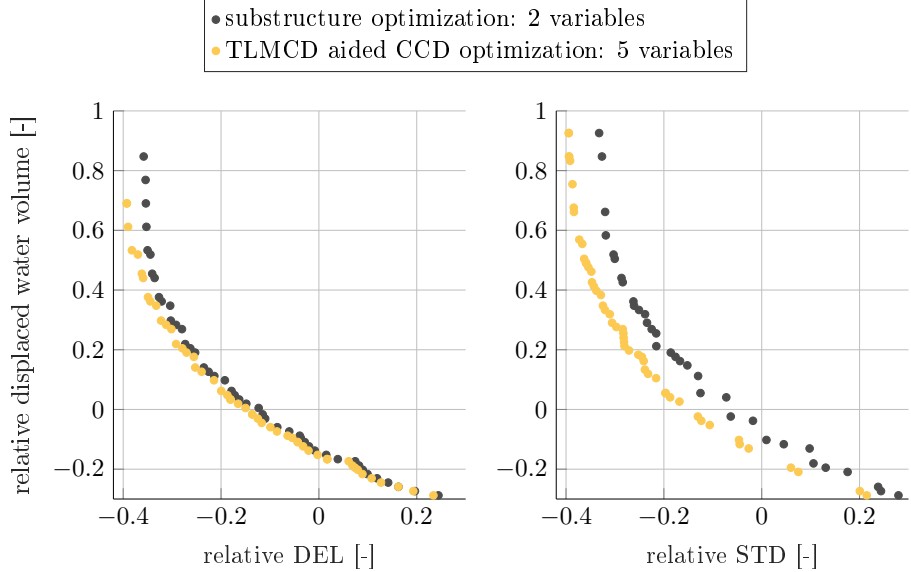

**Figure 8.** Pareto fronts resulting from the 2-variable substructure optimization, showing the trade-off between the relative displaced water volume and the relative costs defined in Equation 2. Left: relative displaced water volume versus relative DEL-cost; Right: relative displaced water volume versus relative STD-cost.

### 4.2.1 Two-objective optimization

The results of a two-objective optimization are presented in Figure 8, with two plots showing the Pareto fronts for different objectives. On the left plot, the objectives are the non-dimensional displacement, which measures the relative displaced volume compared to the original NAUTILUS-10 design, and the relative DEL. On the right plot, the relative DEL objective is changed

to relative STD. The DEL objective, as defined in Equation 2, is a combination of the tower base fore-aft bending moment and the fairlead tension, while the STD objective includes the platform pitch motion and tower top acceleration. For simplicity, these objectives are referred to as DEL-cost and STD-cost in the following discussion. In addition, the change of Pareto front as the size of free variables is increased can be observed by comparing the black and yellow dots.

Looking at the black dots on the plots where only two variables related to the substructure geometry are optimized, two

main observations can be made. Firstly, there is a strong inverse correlation between the DEL-cost/STD-cost and the relative displacement (displaced water volume). This correlation is nearly linear and is evident in the range where the relative displacement is approximately $20\,\%$ above and below zero. While decreasing the displacement can lead to a reduction in the total material, construction, transportation, and installation costs, this always comes at the expense of higher DEL and STD. It is worth noting that the DEL-cost and STD-cost are only slightly reduced when the relative displacement reaches 0.5, indicat-

ing a $50\,\%$ increase in displacement compared to the NAUTILUS-10. The second observation is that the point $(0,0)$, which represents the same cost as the original NAUTILUS-10 design, almost coincides with the Pareto front when optimizing both displacement and STD-cost. This indicates that the NAUTILUS-10 design is one of the optimal choices when considering the



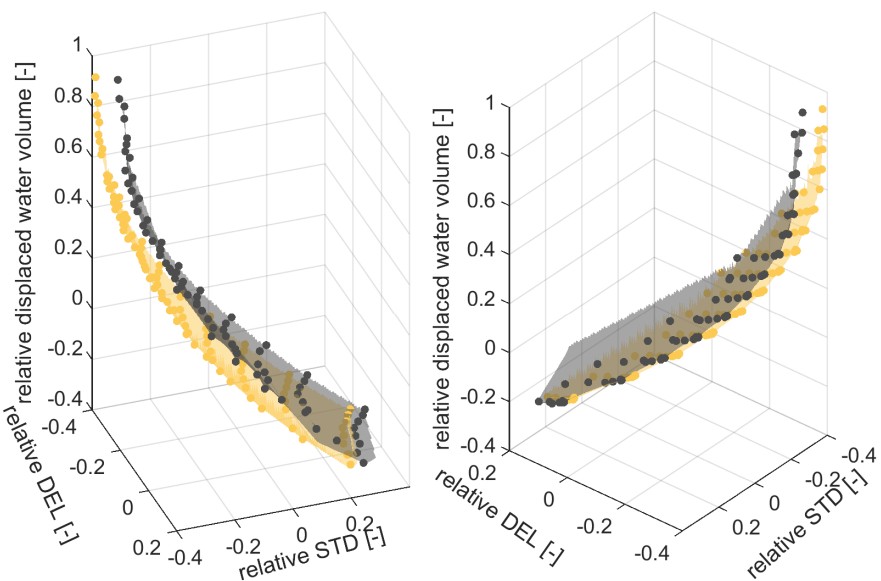

**Figure 9.** Comparison of Pareto optimal surfaces between the two-variable geometric optimization (in black) and the five-variable TLMCD aided CCD optimization (in yellow).

STD-cost. However, this is not the case when the DEL-cost is taken into account. In the left plot, which shows the relationship between DEL-cost and displacement, it can be seen that the point $(0,0)$ lies to the right of the Pareto front, indicating that the

optimal solutions found have better performance than the original design.

The impact of the TLMCD and the blade pitch controller on the substructure geometry optimization can be seen by comparing the yellow and black dots. Obviously, the Pareto fronts are shifted to the left side, indicating an improved overall dynamic response. Despite this, the shape of the Pareto front is very similar to the one obtained from pure substructure geometry optimization, therefore the previous observations still apply. However, a notable improvement in the platform pitch motion can be

found, as the STD-cost is reduced by 5 to $10\,\%$ for the same displacement. In terms of the DEL-cost, the contributions of the TLMCD and the blade pitch controller are limited when the displacement is relatively small. These designs are typically lighter and have higher natural frequencies, which are closer to the wave frequency range. This makes them more susceptible to wave induced excitation, and the dynamic response cannot be significantly improved even with additional damping. Of course, the mass of the TLMCD also plays an important role. As smaller substructures have a TLMCD with less fluid mass, their ability

to compensate for motion induced by aerodynamics is limited. This explains why the designs with larger displacement in the optimization process can achieve a better improvement by including the TLMCD.

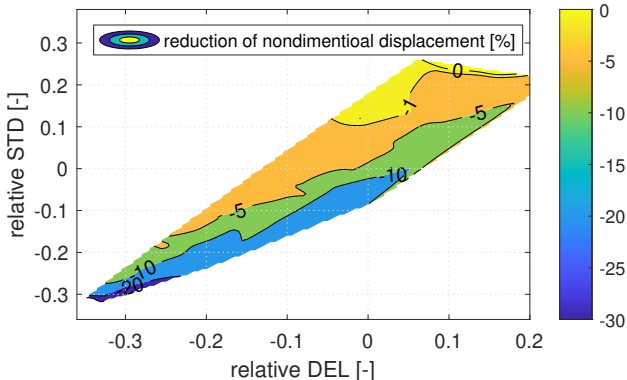

**Figure 10.** Relative reduction in displacement contributed by the TLMCD aided CCD optimization.

### 4.2.2 Three-objective optimization

The Pareto front obtained from a two-objective optimization only shows the optimal solutions for each optimization case, with only two objectives being optimized at a time. However, it is important to note that the decision space may be different for each
of the cases shown in Figure 8, which means that a design that minimizes STD-cost may not necessarily minimize DEL-cost at the same time. To address this limitation, it is necessary to optimize all three objectives.

Figure 9 illustrates the Pareto surfaces from two different views, providing insight into the impact of additional optimization variables and objectives. The main findings from the two-objective optimization are supported by the Pareto surfaces depicted in Figure 9 as well. The inclusion of a TLMCD in the floater design can reduce the displacement required to achieve the same
DEL- and STD-cost, resulting in a lower total substructure cost. However, it is noted that the displacement reduction is most apparent when the STD-cost is relatively low, and when the STD-cost is more than 20 % higher than the original NAUTILUS design, the effect of the TLMCD is almost negligible. This is probably due to the highly dynamic nature of the system, which is strongly influenced by wind and waves, making the use of a TLMCD unhelpful.

To gain a better understanding of the displacement reduction contributed by the TLMCD, the contour lines of the displace-
ment reduction over the STD-cost and DEL-cost are presented in Figure 10. The displacement reduction is defined as the difference in the relative displaced water volume on the Pareto surfaces for the same DEL-cost and STD-cost as shown in Figure 9. A negative value indicates that the design with TLMCD requires less displacement to achieve the same DEL-cost and STD-cost, resulting in cost savings (indirectly reflected by the reduced displacement) without compromising the loads and motions. The results show that a well-designed TLMCD, along with a tailored blade pitch controller, can reduce the FOWT
displacement by up to 20 %. When targeting the DEL-cost and STD-cost of the original NAUTILUS-10, the displacement reduction is approximately in the range of 5 % to 10 %.

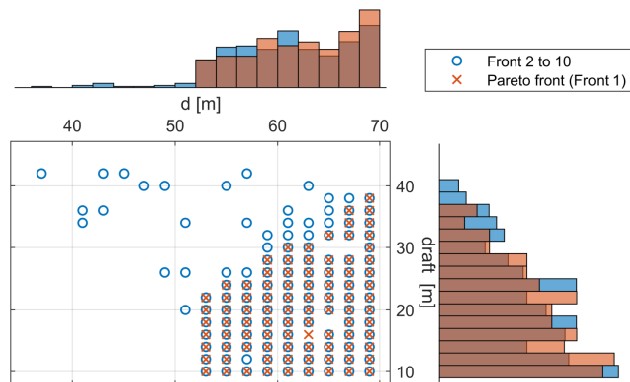

**Figure 11.** Geometric decision space on the Pareto surface.

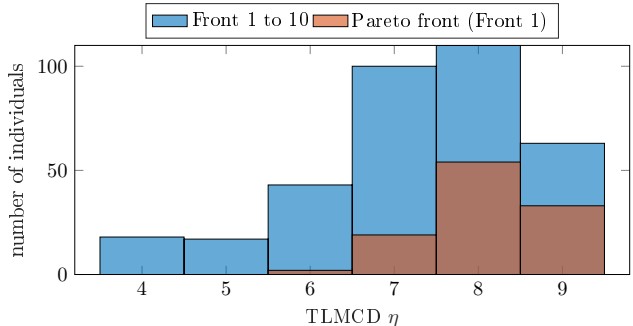

**Figure 12.** Optimal head loss $\eta$ of the TLMCD for the decision space.

## 4.3 Optimal decision space

While the focus so far has been on achieving optimal objectives, it is also essential for system designers to examine the design choices that can ensure good performance. The following section discusses the decision space, which represents the

optimal subsystem designs chosen by the optimiser. This will provide a better insight into how design choices affect the overall performance of the system.

Figure 11 shows the optimal geometric design space of the substructure along with its corresponding histograms. The data indicate that the optimal designs tend to have a larger column spacing $d$, as no solutions are found for $d < 50\,\mathrm{m}$. Designs with column spacing $d \in [66\,\mathrm{m}, 69\,\mathrm{m}]$ account for the highest percentage of designs on the Pareto surface. This observation can be

explained by the increased second order moment of inertia of the water plane area, resulting from the larger column spacing. For the same mass and displacement, a substructure with a larger column spacing generates a higher restoring moment in roll and pitch directions, which reduces the pitch and roll motions. However, it is important to note that the optimization process does not consider the structural integrity of the deck and heave plate, which connect the four vertical columns. Consequently, the distribution of optimal designs may vary if this factor is taken into account. Nevertheless, many optimal solutions are found





slightly below a draft of $60\,\mathrm{m}$, despite the smaller column spacing in this range, which may indicate the presence of local optima.

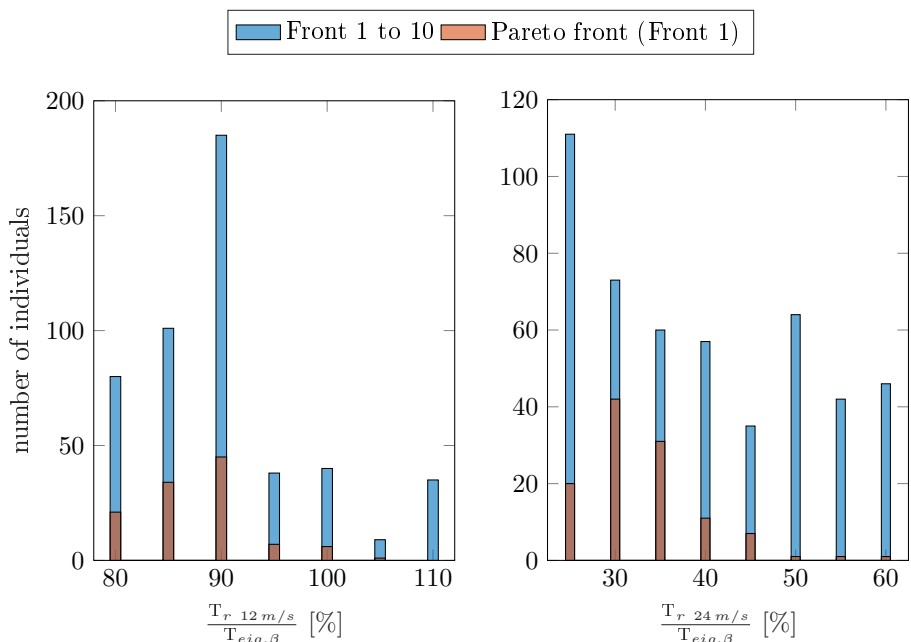

**Figure 13.** Controller decision space on the Pareto surface.

The design of the TLMCD involves only one free variable, namely the head loss $\eta$. The histogram in Figure 12 indicates that most optimal solutions have head loss values ranging from 7 to 9, with $\eta = 8$ being the most frequently selected value. However, it is important to note that this optimal value of $\eta$ is only applicable to this specific concept and defined objectives.

The optimal value of $\eta$ may differ for other designs with different design objectives.

The last two design variables are associated with the blade pitch controller, specifically the rise time $T_r$ of the closed control loop shaping at wind speeds of $12\,\mathrm{m/s}$ and $24\,\mathrm{m/s}$, respectively. It was suggested in (Yu et al., 2020) that the $T_r$ value at $12\,\mathrm{m/s}$ should be slightly smaller than the platform pitch natural period. This recommendation is supported by the optimal $T_{r,12m/s}$ distribution, where $90\,\%$ of the platform pitch natural period is the preferred choice, although a slightly smaller value

of $T_r$ (i.e., $85\,\%$ of the platform pitch natural period) can still achieve satisfactory performance. Regarding the optimal $T_r$ at $24\,\mathrm{m/s}$, the histogram suggests that a value of $30\,\%$ or slightly higher (i.e., $35\,\%$) provides the best performance. It is important to note that these optimal values are specific to the current design and may vary in other wind turbine systems.

## 5 Dynamic Responses of the Optimal Designs

The results of the TLMCD aided CCD optimization show that the optimizer can potentially reduce the displaced tonnage of

a FOWT by up to $20\,\%$. While the optimizer only considers the defined design objectives, it is still important to examine



**Table 3.** Properties and costs of the two selected designs on the optimal Pareto surface.

|  | $d$ | draft | $\eta$ | $T_{r,12}$ | $T_{r,24}$ | $J_1$ | $J_2$ | $J_3$ |
|---|---|---|---|---|---|---|---|---|
|  | [m] | [m] | [-] | [%] | [%] | [%] | [%] | [%] |
| Design-1 | 55 | 14 | 8 | 90 | 30 | -11.6 | -0.73 | -4.5 |
| Design-2 | 61 | 12 | 9 | 105 | 35 | -15.2 | -0.26 | 7.1 |

the dynamic responses of the optimal designs. Hence, two designs with similar DEL-costs and STD-costs to the original NAUTILUS-10 design are selected. The corresponding design space and decision space are listed in Table 3.

Design-1 and Design-2 are two optimal designs on the Pareto surface that are selected for further analysis. Design-1 has a column spacing that is similar to the original NAUTILUS-10, but its draft is four meters shorter, resulting in a reduction in the displaced tonnage of the FOWT by $11.6\%$. In contrast, Design-2 has a larger column spacing and a further reduced draft, resulting in a reduction in the displaced tonnage of $20\%$. Both designs have very similar DEL-costs to the original NAUTILUS-10 design (less than $1\%$ difference). Additionally, Design-1 has a lower STD-cost, while Design-2 has a $7.1\%$ increase in STD-cost, which is a cost for the significantly reduced displacement.

In Figure 14, a statistical analysis of all sensors used in the cost model is presented. The DELs of the tower base bending moment and fairlead tension are quite similar between Design-1 and the original NAUTILUS-10 design, with a slightly lower fairlead tension for the former in the wind speed range of $11\,\mathrm{m/s}$ to $20\,\mathrm{m/s}$. The fairlead tension of Design-2 is further reduced, while the tower base bending moment is increased, resulting in an overall increase in DEL-cost of $7.1\%$. In terms of STD-cost, the nacelle acceleration of the optimal design is higher across all operating wind speeds, mainly due to the reduced draft. This reduction results in a higher overall center of gravity and a lower pitch stiffness, leading to a slightly larger mean platform pitch angle for the optimal designs. At below rated wind speeds, the platform pitch of both designs is similar as the blade pitch is not yet activated. At higher wind speeds, the redesigned blade pitch controller and the positive contribution of the TLMCD significantly reduce the motions. Despite the higher nacelle acceleration, the overall STD-cost of the optimal designs is comparable to that of the original NAUTILUS-10 design since the cost model considers tower top acceleration and platform pitch motion equally in the cost calculation.

Figure 15 provides a detailed comparison of PSD of the dynamic responses at a wind speed of $13.9\,\mathrm{m/s}$. Due to the significant differences in the PSD amplitudes across the frequency range, the plots are split into two parts, with the left plot showing the frequency range from $0\,\mathrm{Hz}$ to $0.05\,\mathrm{Hz}$ and the right plot showing the frequency range from $0.05\,\mathrm{Hz}$ to $0.2\,\mathrm{Hz}$. In the lower frequency range, wind excitation dominates the responses. In addition, the natural frequencies of platform motions also contribute to resonances in this region. Interestingly, Design-1 and Design-2 exhibit significantly lower roll and pitch response amplitudes near $0.03\,\mathrm{Hz}$, which indicates the significant contribution of the TLMCD. This reduction results in lower rotor effective turbulence, which reduces the blade pitch activity, but also the rotor speed oscillation. In the higher frequency range, the system is primarily excited by the waves. However, the amplitudes of the PSDs are generally much smaller in this region. Both Design-1 and Design-2 show higher roll and pitch motions in this range due to the reduced displacement of the

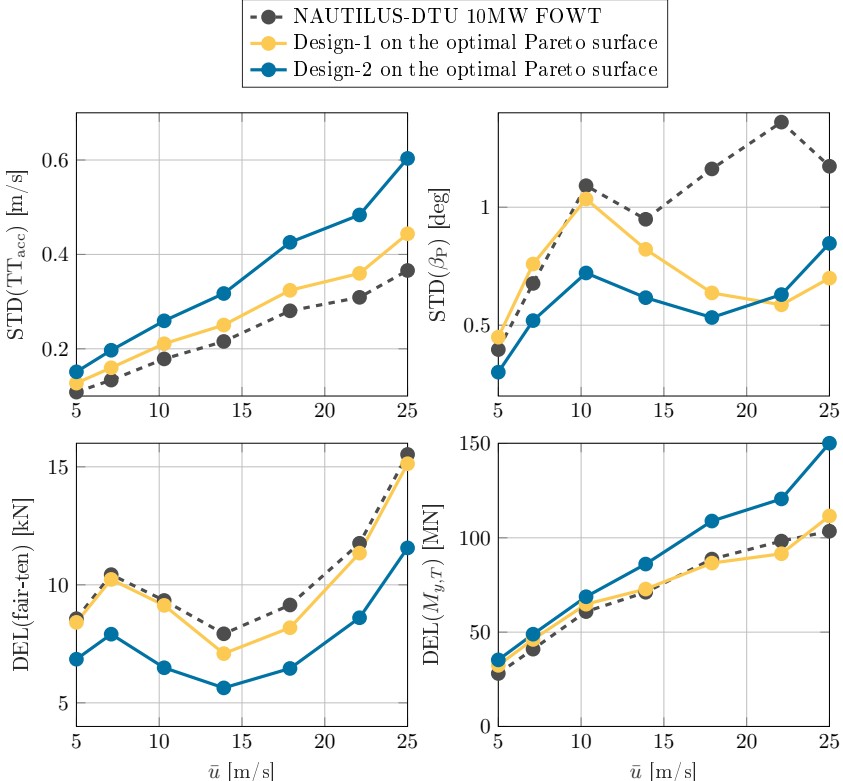

**Figure 14.** Comparison of statistical responses between the original NAUTILUS-10 design and two selected designs with similar STD-cost and DEL-cost on the optimal Pareto surface.

substructure, which negatively affects the dynamic responses in waves. However, the roll and pitch motions of both designs are
410   still slightly lower than those of the NAUTILUS-10 when considering the response across the entire frequency range.

## 6   Conclusions

Integrating a tuned liquid damping system into a FOWT presents a substantial system optimization challenge. The potential benefits of such a damping system to the overall system depend largely on its interaction with the turbine control system, the design of which is also linked to the substructure geometry. This is due to the physical coupling between aerodynamics and
415   hydrodynamics of all three subsystems. As a result, the design of each subsystem has a significant impact on the overall system behavior, and the optimization of these individual subsystems, as well as their efficient cooperation, essentially influence the final LCOE for the FOWT.





**Figure 15.** Comparison of frequency responses between the original NAUTILUS-10 design and two selected designs with similar STD-cost and DEL-cost on the optimal Pareto surface at wind speed $13.9\,\mathrm{m/s}$.

In this context, a comprehensive multi-objective CCD optimization framework has been developed to optimize the substructure geometry, the TLMCD, and the blade pitch controller, systematically incorporating a tuned liquid damping system into a
420   FOWT. To address the challenge posed by the coupling, the framework explores the design space of all three subsystems simul-



taneously, searching for the optimal synergy between them to achieve a good balance between production cost and response performance.

Based on the case study using the Lifes50+ NAUTILUS-10 design, a well-designed TLMCD can reduce both FOWT motions and loads. In addition, the blade pitch controller has more flexibility as the negative aerodynamic damping is partially compensated by the additional damping of the TLMCD. This also helps to reduce the power fluctuations. On the other hand, if the motions and loads can remain similar to the reference design, a much lighter substructure design can be achieved. This contributes to reducing the manufacturing costs without deteriorating the motion and load performance. In summary, these results show that there is a significant potential to improve the LCOE of a FOWT by incorporating a TLMCD and adapting the controller.

*Author contributions.*

Wei Yu: Conceptualization, Methodology, Implementation of the framework, Analysis, Writing - Original draft preparation. Sheng Tao Zhou: Parameterization of the NAUTILUS-10 Semi-Sub substructure, Writing - Reviewing and Editing. Frank Lemmer: Supervision, Writing - Reviewing and Editing. Po Wen Cheng: Supervision, Writing - Reviewing and Editing.

*Competing interests.*

The authors declare that there are no competing interests associated with this research.

*Acknowledgements.* This research was partly supported by the CROWN project. The CROWN project has received funding from the Eurostars-2 joint programme with co-funding from the European Union Horizon 2020 research and innovation programme.



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
