# Peer review of "Control Co-Design optimization of floating offshore wind turbines with tuned liquid multi-column dampers"

_Wind Energy Science, 2023_

## Author Comment (AC1)

**RC1**

The following are the basic comments on "Control Co-Design optimization of floating offshore wind turbines with tuned liquid multi-column dampers": the text is logically rigorous; the main lines are clear, and the science is strong. The basic opinion on "Control Co-Design optimization of floating offshore wind turbines with tuned liquid multi-column dampers" is as follows: The whole paper is logically rigorous, with a clear main line and a strong scientific basis. The topic is novel and of practical significance to the project, which is the main reason for the acceptance of the proposal. However, the paper still has the following problems that need to be explained and corrected:

1. The article clearly describes the background and importance of the study but does not discuss in detail the literature survey of related work. The authors should have provided a more comprehensive overview of the relevant literature so that the reader can understand the current state of the research.

We will extend the literature review on TLCD/TLMCD in the final version.

2. while an overview of the multi-objective CCD optimization framework is provided, specific technical details and algorithm selection seem to be missing. To enhance transparency and reproducibility, it is recommended that more methodological details be provided.

The optimizer NSGA-II (Non-dominated Sorting Genetic Algorithm II) is a very classical multi-objective optimizer and has been widely used also in wind energy research, so we only have a short introduction in Section 3.2. Since we use an existing Matlab toolbox for the optimizer, we will add the citation of this Matlab toolbox for transparency and reproducibility.

3. In some places, the interpretation of the data seems to be less than intuitive, which may lead to reader confusion about some of the conclusions. A more in-depth explanation of key data and graphs is recommended.

We apologize for the lack of clarity in the data and figures. Sometimes it is difficult for us to identify the depth of the explanation as we are so familiar with everything in the manuscript. We would be grateful if the reviewer could point out the exact table number and figure number, where we should provide more detailed explanations. We will add the information accordingly in the final version.

4. The article mentions that the optimization process did not consider the structural integrity of the deck and heave plate. This is an important omission that may affect the effectiveness of the optimization. It is recommended that at least some preliminary structural analysis be performed to confirm the feasibility of the optimized design.

This manuscript focuses on the methodology development rather than providing an optimal solution. The result shows the impact of TLMCD assisted CCD optimization on the dynamic performance of the system, so we are also not trying to identify the optimal design. Instead, we only take two examples on the Pareto surface to demonstrate the dynamic performance, including one example which has the same column spacing as the original design and should not have the

structural integrity problem due to the increased column spacing. For realistic industrial design, the structural integrity of the designs can be further analyzed and eliminated from the resulting Pareto surface. This however does not affect the methodology in this study. Therefore, instead of adding additional structural analysis, we would highlight this topic as an outlook in the final summary.

5. Although the optimization of the TLMCD is mentioned, there seems to be a lack of a detailed description of its specific operation principle and the selection of design parameters. It is suggested that more information on the design considerations of the TLMCD be provided.

We have described the design principles of the TLMCD between Line170 to Line180, in addition, Line267 to Line271describes the implementation of the design process in the optimization framework. We also highlighted there that the method has already been published in (Yu et al., 2019), so we didn't repeat the same content in this manuscript to keep concise. If the reviewer think there is something essential which needs to be emphasized again, we could add more content on this topic.

6. The article mentions the optimization of the blade pitch controller but does not discuss other possible control strategies or methods in detail. Considering the importance of control strategies to FOWT performance, a more comprehensive control strategy discussion is recommended.

Similar to the TLMCD design, the parametric design of the controller has also been published in (Yu et al., 2020), where a comprehensive methodology description can be found, therefore we didn't include too much information. However, in response to the reviewer's comment, we will add another paragraph in Section 3.5.2 to summarize the work of (Yu et al., 2020), so that readers can follow more about the controller design in the optimization framework.

7. When listing the optimization variables, it is recommended that a detailed explanation be provided as to why these variables were selected and how they affect the system performance.

Together with the design principles of the subsystem designs, we derived the free variables in Section 3.1. As mentioned, one of the major constraints is the computational costs when selecting design variables. So we select only one or two most important design variables for each subsystem. Of course, if the computational resources allow, more free variables can be selected, such as the diameter of the floater columns, which is also an influencing parameter. However, this required only minor modifications inside the structural design block of the optimization framework. As this manuscript focuses on demonstrating a working methodology, rather than providing an optimal design, we kept only five free variables, and we think the description in Section 3.1 already justify our selection.

8. Comparisons of dynamic response appear to be based on specific wind speeds. It is recommended that a wider range of wind speeds be provided to evaluate the performance of the optimized design more fully.

Figure 14 shows a comparison of the statistics of the system responses for different wind speeds, illustrating the performance of two designs found on the Pareto surface. To limit the length of the manuscript, we think it is sufficient to show the PSD of only one load case of these wind speeds. Only if the reviewer thinks it is critical, we are able to add PSD plots for another wind speed.

9. The topic of the article is to improve the LCOE of FOWTs, but there seems to be a lack of detail on how LCOE is calculated or evaluated. It is recommended that more details on LCOE evaluation be provided.

As stated in the manuscript, as an academic research, we are not able to calculate the LCOE reliably as it is largely influenced by the industry and the market and can vary significantly from time to time, such as material costs. As a consequence, using inaccurate LCOE as an optimization objective may mislead the optimization path. So we selected several representative physical indicators which can reflect indirectly the LCOE to demonstrate the physical benefits of applying CCD-optimization including a TLMCD.

10. The conclusion section appears to be a straightforward restatement of the study results. It is recommended that more in-depth engineering insights and directions for future work be provided in the conclusion.

We will extend the final summary by adding more engineering insights and outlook, in particular, we will highlight that the structure analysis will be added into the optimization for future work.

**RC2**

This paper proposes a control of co-design optimization scheme for FOWT by considering the blade pitch conrol and TLMCD. The paper is well structured and the process is well developed. However, i would recommend the acceptance of the paper if they could address the following issues:

1. The effectiveness of the controller and TLMCD is presented for wind speed larger than rated wind speed. How is the situation for the wind speed is lower than the rated wind?

The interaction between the controller and the platform dynamics is mainly at above rated operating points, as the blade pitch controller affects the aerodynamic thrust, which causes additional loading in platform pitch or roll (depending on the turbine yaw position). This is also one of the main challenge in designing the controller for FOWT, i.e. the instability caused by negative aerodynamic damping and this is also the problem we want to address. At below rated wind speeds, as the generator torque controller only changes the aerodynamic torque which does not interact with the platform motions, the effect of the TLMCD is similar to an anti-roll tank in the ship industry. Nevertheless, we have compared the response statistics of two optimal designs on the resulting Pareto surface to illustrate the impact of the TLMCD and Control Co-Design on the system performances in Figure 14. We believe that additional PSD plots for below rated wind speed may not be essential to draw the conclusions of this work.

2. The repeated word damper is used in the abstract.

Yes, this was a typo, we will fix it in the final version.

3. What is the feasibility if the wind and wave direction is different from the x-axis?

This is actually an advantage of the TLMCD. Compared to installing multiple conventional TLCDs, the TLMCD can damp both roll and pitch directions of the platform and is more robust. One publication (Coudurier et al., 2018) we have cited has a detailed discussion on this topic, since the focus of this work is not on this, we will omit this discussion, but we will highlight this in the introduction part by referring to that publication.

4. In the industry, how this TLMCD can be implemented on a real platform?

Tuned liquid column dampers (TLCDs), which is also known as anti-roll tanks in the ship industry, have been applied commercially for many decades. The construction, installation and maintenance of this type of damper is technically mature. Although we have increased the number of vertical columns, we believe that if the design of a TLMCD for a FOWT is proven, the implementation should not be a barrier.

5. For the platform optimization, the column diameter is a key parameter that can affect the total mass significantly. I recommend adding this parameter to the variables for platform optimization.

Since the paper focuses on the methodology how we can include TLMCD and control co-design in the optimization loop, we compared the case where only the platform is optimized and where both platform and TLMCD assisted Control Co-Design (CCD) is implemented, which can address the added value introduced by the TLMCD and control CCD. As adding additional free variables to the platform increases significantly the computational requirements, we would leave this part for future work where the focus is to reach a realistic final design. We will also highlight this discussion in the summary again.

6. What is the additional cost for this TLMCD used for 10 MW wind turbine?

As for academic research, the actual cost of the TLMCD will be unknown, and we also believe that it will vary significantly in different markets and countries, as well as the amount of production etc. As an early-stage conceptual level research, we focus on the system dynamic performance, rather than the economic aspects, which is also the reason why we only add physical indicators in the cost functions.

7. Compared with the original design, additional TLMCD will change the COM and draft of the platform. Do you consider these differences when you set different models?

The mass ratio of the TLMCD, limited to 3% of the total FOWT mass, is relatively small. In addition, the TLMCD is installed at the keel, where ballast water should be installed, so we don't change the COM of the FOWT, but only adjust the overall mass to maintain the original design draft. We have validated our numerical model in another publication (Yu et al., 2023), where we can match the decay tests very well, so we have kept this simplification.